# Concise Multi-head Attention Models

## Abstract

Attention based Transformer architecture has enabled significant advances in the field of natural language processing. In addition to new pre-training techniques, recent improvements crucially rely on working with a relatively larger embedding dimension for tokens. Unfortunately, this leads to models that are prohibitively large to be employed in the downstream tasks. In this paper we identify one of the important factors contributing to the large embedding size requirement. In particular, our analysis highlights that the scaling between the number of heads and the size of each head in the existing architectures gives rise to this limitation, which we further validate with our experiments. As a solution, we propose a new way to set the projection size in attention heads that allows us to train models with a relatively smaller embedding dimension, without sacrificing the performance.

## 1 Introduction

Attention based architectures, such as Transformers, have been effective for sequence modelling tasks such as machine translation (Gehring et al., 2017; Vaswani et al., 2017), question answering, sentence classification (Radford et al., 2018; Devlin et al., 2018) and document generation (Liu et al., 2018). These models have emerged as better alternatives to the recurrent models - RNNs (Sutskever et al., 2014), LSTMs (Hochreiter & Schmidhuber, 1997) and GRUs (Cho et al., 2014). This is mainly due to their feed forward structure, which removes the sequential processing bottleneck for sequence data, making them easier to train compared to the recurrent models. Self attention models also have found applications in vision (Wang et al., 2018), adversarial networks (Zhang et al., 2018), reinforcement learning (Zambaldi et al., 2018; Li, 2017) and speech recognition (Chiu et al., 2018).

Recent advances in using the self attention models in natural language tasks have been made by first using a language modeling task to pre-train the models and then fine tuning the learned models on specific downstream tasks. Radford et al. (2018) and Devlin et al. (2018) used Transformers to pre-train a language model and showed that the fine tuned model outperforms LSTMs on many natural language understanding and question answering tasks. For example, BERT (Devlin et al., 2018), a 24 layer transformer model, is shown to achieve the state of the art performance on several NLP tasks, including on the SQuAD dataset. These advances, in addition to novel pre-training tasks, relied on bigger models with a larger embedding size. BERT model uses an embedding size of 1024 (Devlin et al., 2018); GPT-2 uses models with embedding size up to 1600 (Radford et al., 2019).

A single Transformer block consists of two key components: a multi-head self attention layer followed by a feed forward layer (Vaswani et al., 2017). A single head in a multi-head attention layer, computes self attention between the tokens in the input sequence, which it then uses to compute a weighted average of embeddings for each token. To keep the number of parameters fixed in the attention layer, each head projects the data into a lower dimensional subspace, dimension of which scales as 1/(number of heads), and computes the self attention in this subspace. This projection size for each head is commonly known as the *head size*.

Despite the advances in using Transformer models for various tasks, their functioning and design choices still remain mysterious and are not well understood. Can the attention layer learn arbitrary contextual representations? What is the role of the feed forward layer in the Transformer block? Do we need such a large embedding size to capture the context of all the tokens? Answering these questions requires an understanding of the representation power of the units in the Transformer.

In this paper we take some of the first steps towards developing such an understanding of the Transformer. In particular, we focus on the representation power of the multi-head self attention layer.

| # heads | 8 | 16 | 32 |
|---|---|---|---|
| # params | 336M | 336M | 336M |
| SQuAD - F1 | 90.89±0.15 | 90.61±0.14 | 90.45±0.08 |
| SQuAD - EM | 84.1±0.34 | 83.75±0.27 | 83.48±0.13 |
| MNLI | 85±0.2 | 84.5±0.4 | 84.4±0.2 |

Table 1: Performance of BERT$_{LARGE}$ (Devlin et al., 2018), a 24 layer Transformer with an embedding size of 1024, does not improve with the increasing number of heads after 8 heads.

First, we analyze the representation power of a single self attention unit and show that it crucially depends on the projection sizes used to compute the dot product attention.

We next study the advantage of having multiple heads in the attention layer. It is generally believed that increasing the number of heads helps by allowing the heads to compute context from different representation subspaces at different positions. However, increasing the number of heads decreases the head size, decreasing the expressive power of individual heads. We show that reducing the head size to a value below the input sequence length hurts the representation power of each head. This is because a smaller head size introduces a rank constraint on the projection matrices in each head, and limits their representation power. We indeed notice this effect in practice: while the performance improves with increasing the number of heads in the beginning (Devlin et al., 2018), we notice a drop in the performance once the number of heads increases beyond a certain threshold, as seen in Table 1 and Fig. 1 (see also Table 4(A) in Vaswani et al. (2017)).

This heuristic of scaling the head size inversely with the number of heads was proposed initially in Vaswani et al. (2017) and has become the standard way of using multi-head attention (Radford et al., 2018; Devlin et al., 2018). In order to avoid hurting the performance, the existing Transformer models allow for multiple heads by increasing the embedding size, which in turn increases the head size. However, larger embedding size, in addition to increasing the number of parameters, makes it expensive to use the model and the learned embeddings in downstream tasks, as the downstream model sizes scale with the embedding size of the tokens. For example, the inference time and memory required in retrieval tasks increases linearly with the embedding size.

Based on these observations, we propose a new way to set the projection size in the attention heads, in which each head has a fixed head size that is independent of both the number of heads and the embedding size of the model. This allows us to train models with a relatively smaller embedding size without affecting the head size. It also allows us to increase the number of heads per layer to improve the performance. Another advantage of the fixed head size Transformer is, unlike the standard Transformer, which requires the number of heads to be a factor of the embedding size, we are free to set arbitrary number of heads as required for the task.

We evaluate this fixed head size Transformer on language modeling (LM1B dataset), natural language inference (MNLI dataset) and question answering tasks (SQuAD dataset). We show that the modified Transformer trained with an embedding size of 512 can match the performance of the BERT$_{LARGE}$(Devlin et al., 2018), a Transformer with an embedding size of 1024 (see Fig. 2). We further present experimental results evaluating the effect of different choices of the head size and the embedding size in the Section 4.

The contributions of this paper are summarized below.

- We analyze the representation power of the multi-head self attention layer and show the limitation the embedding size places on the number of heads.

- We propose a new way to set the head size, and show the proposed fixed head size layers are strictly better than the standard multi-head attention layers in terms of their expressive power. This modification allows us to both increase the number of heads per layer and decrease the embedding size, without hurting the performance.

- We experimentally show that the fixed head size Transformer can be trained with a smaller embedding size and more heads on three standard NLP tasks.

## 1.1 Related Works

Given the significance of self attention models, there has been work trying to both improve the performance and speedup the computation in Transformers. Ott et al. (2018) and You et al. (2019) reduce precision and use large batch training to reduce the training time of the attention models. Child et al. (2019) propose sparse self attention models to speed up the computation in the attention layer for long sequence data generation tasks. They show that these sparse attention models can be trained on tasks with sequence length greater than 10k without sacrificing the accuracy. Dehghani et al. (2018) propose a depth recurrent Transformer network that reuses the parameters across layers. They show that this modification makes the Transformer networks Turing complete even with finite precision weights. Yang et al. (2019) propose a new way to increase the effective sequence length that the Transformer attends to, by reusing the intermediate embeddings across sequences. They show that the modified architecture performs better on tasks that require computing context over longer sequence lengths. We note that most of these modifications rely on the multi-head self attention, the same building block of the Transformers. Our work is studying this basic multi-head attention layer, and suggesting a new way to set the head size, which can be easily used along with any of the above architectural modifications.

Wu et al. (2019) propose to replace the self-attention layer with lightweight dynamic convolutions and show improved performance on machine translation and language modeling. Even though the resulting model has faster inference time, it still needs to use a large embedding size (1024), as big as the original attention models. We believe the techniques in this paper can be combined with these results to realize both smaller embedding size and faster inference time.

Sun et al. (2019) perform neural architecture search using evolutionary methods on sequence to sequence models and find an evolved transformer architecture, which in addition to multi-head attention units, has convolution filter and gated linear units. Our proposed modifications stay closer to Transformers in spirit and can be used as seed units for this architecture search.

Voita et al. (2019); Michel et al. (2019) study the importance of different heads in an attention layer. They observe that, during inference, many of the heads in each layer can be pruned away with a little effect on the prediction. However, they still need multiple heads during the training.

Child et al. (2019); Correia et al. (2019) impose sparsity structure on the attention layer during training to improve both interpretability and performance. Fixing the head size will in fact make it easier to learn such sparsity patterns, as a low rank constraint does not allow a head to express all possible sparsity patterns. Combining these techniques can hence potentially enable training of sparse attention models with a smaller embedding size.

## 2 Transformer Architecture and Analysis

In this section we present the Transformer architecture and analyze the representation power of the multi-head self attention, a key component of the Transformer block.

The input to a Transformer network is a sequence of $n$ tokens. Typically, each token is converted into a token embedding of dimension $d$ by an embedding layer. We let $\mathbf{X} \in \mathbb{R}^{d \times n}$ be the embedding matrix corresponding to the $n$ tokens in the input sequence.

### 2.1 Single-Head Attention

The transformer block is a combination of a self attention layer followed by a feed forward layer (Vaswani et al., 2017). Both layers have a skip connection and use Layer Normalization (LN) (Ba et al., 2016). In particular, for token embeddings $\mathbf{X}$, the dot product attention is computed as follows.

$$\text{Attention}(\mathbf{X}) = \mathbf{W}_v \mathbf{X} \cdot \text{Softmax}\left[\frac{(\mathbf{W}_k \mathbf{X})^T (\mathbf{W}_q \mathbf{X})}{\sqrt{d_k}}\right] = \mathbf{W}_v \mathbf{X} \cdot \mathbf{P}. \tag{1}$$

Here $\mathbf{W}_q \in \mathbb{R}^{d_q \times d}$, $\mathbf{W}_k \in \mathbb{R}^{d_k \times d}$ and $\mathbf{W}_v \in \mathbb{R}^{d_v \times d}$ represent the projection matrices associated with the query, key and value respectively in an attention unit (Vaswani et al., 2017). For a single-head attention unit, we have $d_q = d_k = d_v = d$. In the dot-product attention (cf. (1)), $\mathbf{P}$ aims to capture the context of the input for a given token based on the remaining tokens in the input sequence.

Subsequently, the output of the attention layer takes the following form.

$$\text{LN}\left(\mathbf{X} + \mathbf{W}_o \cdot \text{Attention}(\mathbf{X})\right), \tag{2}$$

where $\text{LN}(\cdot)$ represents the layer-normalization operation. Given the attention module, as defined in (1), it is natural to question its ability to represent arbitrary contexts $\mathbf{P}$ for a given input sequence $\mathbf{X}$.

Towards this, we show that for a large enough projection dimension $d$, the unit has enough capacity to represent arbitrary contexts over a given input sequence. In the following result we establish that for a large enough projection size an attention unit can represent any data pair $(\mathbf{X}, \mathbf{P})$. We also show that the model cannot represent arbitrary context when $d$ is smaller than $n$.

**Theorem 1** (Representation Theorem)**.** *If $d_q = d_k = d \geq n$, then given any full column rank matrix $\mathbf{X} \in \mathbb{R}^{d \times n}$ and an arbitrary $n \times n$ positive column stochastic matrix $\mathbf{P}$, there always exists $d \times d$ projection matrices $\mathbf{W}_q$ and $\mathbf{W}_k$ such that*

$$\text{Softmax}\left[\frac{(\mathbf{W}_k\mathbf{X})^T(\mathbf{W}_q\mathbf{X})}{\sqrt{d_k}}\right] = \mathbf{P}. \tag{3}$$

*If $d_q = d_k = d < n$, there exist $\mathbf{X}$ and $\mathbf{P}$ such that (3) does not hold for all $\mathbf{W}_q$ and $\mathbf{W}_k$.*

The proof of Theorem 1 is provided in the supplementary material. This result shows that the projection dimension $d_q = d_k = d$ needs to be larger than the sequence length $n$ for the attention unit to be able to represent any desired context $\mathbf{P}$. Even though this result describes a single example sequence case, it highlights a fundamental property of the model architecture that increasing the projection size increases the capacity of the attention heads.

## 2.2 MULTI-HEAD ATTENTION

As discussed in Section 2.1, an attention unit updates the embedding of an input token based on a weighted average of the embeddings of all the tokens in the sequence, using the context $\mathbf{P}$ (cf. (1)). Vaswani et al. (2017) proposed Multi-Head attention mechanism that increases the representation power of an attention layer, where multiple attention units operate on different low dimensional projections of the input, with each attention unit being referred to as a head. This is followed by concatenation of the outputs from different heads. In particular, the computation inside a Multi-Head attention with $h$ heads takes the following form:

$$\text{head}(\mathbf{X})_i = \mathbf{W}_v^i\mathbf{X} \cdot \text{Softmax}\left[(\mathbf{W}_k^i\mathbf{X})^T(\mathbf{W}_q^i\mathbf{X})/\sqrt{\tfrac{d}{h}}\right] \in \mathbb{R}^{\frac{d}{h} \times n}$$

$$\text{MultiHead}(\mathbf{X}) = \text{Concat}[\text{head}(\mathbf{X})_1, \cdots, \text{head}(\mathbf{X})_h] \in \mathbb{R}^{d \times n}.$$

The output of the Multi-head attention layer then becomes

$$\mathbf{Z} = \text{LN}\left(\mathbf{X} + \mathbf{W}_o \cdot \text{MultiHead}(\mathbf{X})\right), \tag{4}$$

where $\mathbf{W}_o \in \mathbb{R}^{d \times d}$. For a model with $h$ heads, the query, key and value projection matrices $\{\mathbf{W}_q^i\}$, $\{\mathbf{W}_k^i\}$ and $\{\mathbf{W}_v^i\}$ are $\frac{d}{h} \times d$ matrices. Therefore, each head projects the input onto a $\frac{d}{h}$-dimensional subspace to compute the context, and keeps the number of parameters fixed per layer. This has been observed empirically to increase the expressive power of the attention layer (Vaswani et al., 2017).

## 2.3 DEPENDENCE OF THE NUMBER OF HEADS ON THE EMBEDDING SIZE

While increasing the number of heads seemingly gives the model more expressive power, at the same time we are reducing the head size, which can decrease the expressive power. When the number of heads $h$ is larger than $\frac{d}{n}$, the attention unit inside each head projects onto a dimension smaller than $n$, and looses its ability to represent arbitrary context vectors (cf. Theorem 1). Since the sequence length is fixed from the data/task at hand, the only remaining way to increase the number of heads, without loosing the expressiveness, is by increasing the embedding size $d$. This corresponds to a fundamental limitation of the model architecture that one needs to increase the embedding size in order to support more heads.

Unfortunately, increasing the embedding size leads to higher computation and memory requirements to train and store the model. Further, since it is common to use learned embeddings from Transformer based models for downstream tasks (Devlin et al., 2018), larger embedding size increases the model size and computation required for all the downstream tasks as well. Given the widespread success of attention mechanism, this highlights the need for a modified model architecture that can leverage the advantages of MultiHead without suffering from the requirement of large embedding sizes.

# 3 CONCISE MULTI-HEAD TRANSFORMER

In this section we propose a different way of setting the head size of the Transformer, which allows us to enjoy the advantage of higher expressive power of multiple heads without requiring the embedding size to be large. The key is to decouple the dependency between the projection size in a head and the embedding size of the model. The projection matrices now project onto subspaces of a fixed dimension $d_p$ irrespective of the number of heads $h$. This approach where $d_p$ is independent of $d$ and $h$ leads to the following attention mechanism.

$$\text{fixedhead}(\mathbf{X})_i = \mathbf{V}_v^i \mathbf{X} \cdot \text{Softmax}\left[(\mathbf{V}_k^i \mathbf{X})^T (\mathbf{V}_q^i \mathbf{X}) / \sqrt{d_p}\right] \in \mathbb{R}^{d_p \times n}$$

$$\text{FixedMultiHead}(\mathbf{X}) = \text{Concat}[\text{fixedhead}(\mathbf{X})_1, \cdots, \text{fixedhead}(\mathbf{X})_a] \in \mathbb{R}^{d_p \cdot h \times n}.$$

Note that the projection matrices used here $\{\mathbf{V}_q^i\}$, $\{\mathbf{V}_k^i\}$ and $\{\mathbf{V}_v^i\}$ are $d_p \times d$ matrices. With $\mathbf{V}_o \in \mathbb{R}^{d \times h \cdot d_p}$, the output of this new multi-head attention layer takes the following form.

$$\mathbf{Z} = \text{LN}\left(\mathbf{X} + \mathbf{V}_o \cdot \text{FixedMultiHead}(\mathbf{X})\right).$$

This modification makes each attention head more similar to a hidden unit in a feed forward network or a filter in a convolutional network, and allows us to vary the number of heads without the worry of reducing the representation power per head. The downside is, unlike the standard MultiHead, the number of parameters per layer increase with the number of heads. However, this modification allows us to train a model with a smaller embedding size, ultimately allowing us to reduce the total number of parameters in the model.

**Choice of the head size.** Our proposed modification introduces head size $d_p$ as a new model hyper-parameter. We choose head size to be 128 for our BERT experiments, as most of the pre-training is done with 128 sequence length data. While we have ablation studies (cf. Table 2(B)) showing bigger head size improves the performance, there is a tradeoff between increasing the head size vs number of heads vs layers. We found that having sufficient head size matching the pre-training sequence length, is better than having a larger embedding size (cf. Section 4).

## 3.1 MULTIHEAD VS. FIXEDMULTIHEAD ATTENTION

Given a MultiHead layer, we can always represent it using a FixedMultiHead layer, whenever we have the head size $d_p \geq d/h$. While this shows that increasing the number of heads $h$ beyond $d/d_p$ makes individual heads of the FixedMultiHead as expressive as the ones in the MultiHead, it is not clear if FixedMultiHead is *strictly* more expressive. Can the FixedMultiHead layer represent functions that the standard MultiHead layer can not represent? In this subsection we show that indeed, in the multi-head regime, the FixedMultiHead layer is strictly better than the standard MultiHead layer in terms of expressive power.

Consider the standard multi-head attention units in (4).

$$f_{\mathbf{W}}(\mathbf{X}) = \mathbf{W}_o \cdot \text{MultiHead}(\mathbf{X}).$$

We denote the collection of all parameter matrices as $\mathbf{W}$. Similarly, consider the function represented by the fixed head size attention units:

$$g_{\mathbf{V}}(\mathbf{X}) = \mathbf{V}_o \cdot \text{FixedMultiHead}(\mathbf{X}).$$

Let $\mathbf{V}$ be the collection of all these parameter matrices. We define $\mathcal{F}$ and $\mathcal{G}$ to be the class of functions $f_{\mathbf{W}}(\cdot)$ and $g_{\mathbf{V}}(\cdot)$, respectively. As noted above, if $d_p \geq d/h$, we have $\mathcal{F} \subset \mathcal{G}$.

The following theorem shows that even for simple examples in $\mathcal{G}$, functions in $\mathcal{F}$ fail to approximate them beyond certain accuracy; this shows that $\mathcal{F}$ is a *strict* subset of $\mathcal{G}$.

**Theorem 2.** *Given $n \geq 2$ and $d \geq d_p$, let $h > d/d_p$. Consider a fixed head size attention layer $g_{\mathbf{V}}(\cdot)$ with parameters that satisfy the following conditions:*

$$\mathbf{V}_o \times \begin{bmatrix} \mathbf{V}_v^1 \\ \vdots \\ \mathbf{V}_v^h \end{bmatrix} \text{ is full rank, and } (\mathbf{V}_k^i)^T \mathbf{V}_q^i = \mathbf{U} \text{ for all } i = 1, \ldots, h, \text{ where } \mathbf{U} \text{ is a rank-}d_p \text{ matrix.}$$

*Then, for any $f_{\mathbf{W}} \in \mathcal{F}$, there exists $\mathbf{X} \in \mathbb{R}^{d \times n}$ such that $f_{\mathbf{W}}(\mathbf{X}) \neq g_{\mathbf{V}}(\mathbf{X})$.*

Because $\|f_{\mathbf{W}}(\mathbf{X}) - g_{\mathbf{V}}(\mathbf{X})\|$ is a continuous function of $\mathbf{X}$, existence of such an $\mathbf{X}$ implies that the integral of the norm of difference (i.e., approximation error) is strictly positive.

This theorem shows that the expressive power of the FixedMultiHead attention function class is strictly superior to the standard MultiHead attention function class. Hence the heuristic of reducing the head size with the number of heads is limiting the expressive power of MultiHead, whereas using the fixed head size Transformers will increase the expressive power of the attention layers.

## 4 EXPERIMENTS

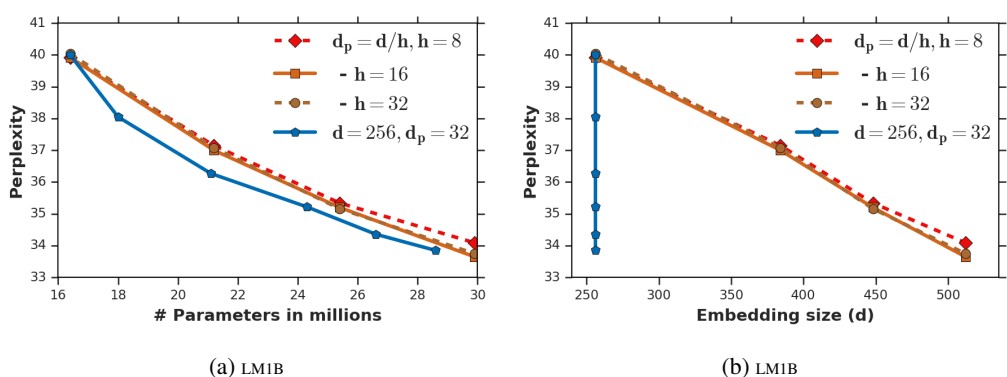

(a) LM1B                                          (b) LM1B

Figure 1: Performance of the standard Transformer ($d_p = {}^d\!/_h$) compared with the fixed head size ($d_p = 32$) models on a language modeling task (LM1B) on the test set. We vary the embedding size of the standard Transformer from 256 to 512. We train the fixed head size models with a fixed embedding size of 256 and a head size of 32, and vary the number of heads from 4 to 70, while matching the number of parameters. The plots clearly indicate that fixing the head size allows us to train Transformers with a smaller embedding size (plot (b)), and with a better scaling of performance (plot (a)). Note that for perplexity lower values are better.

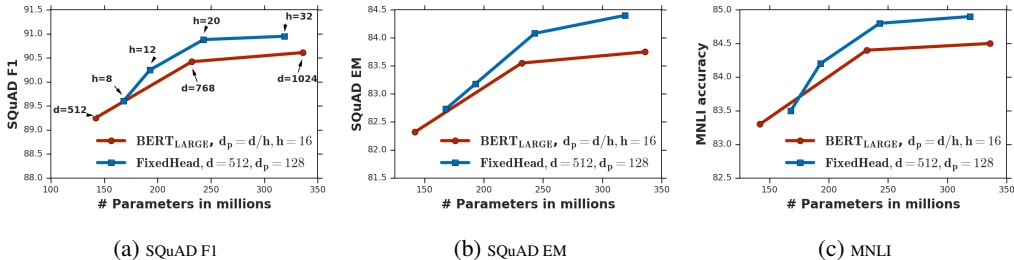

(a) SQuAD F1                        (b) SQuAD EM                        (c) MNLI

Figure 2: Comparison of a 24 layer standard Transformer model BERT$_{\text{LARGE}}$ with the fixed head size model on the SQuAD and MNLI dev sets. We vary the embedding size of the BERT models from 512 to 1024. We train the fixed head size models with a fixed embedding size of 512 and a head size of 128, with a varying number of heads from 8 to 32, while matching the number of parameters. Fixing the head size allows us to train models with an embedding size of 512 with a better performance.

In this section we present our experiments on three standard NLP tasks, language modeling (LM1B), question answering (SQuAD), and sentence entailment (MNLI), to demonstrate: 1) Increasing the number of heads beyond certain point hurts the performance of the standard Transformer, but always helps with our proposed modification; 2) Decoupling the head size from embedding size allows us to train models with a smaller embedding size; and 3) Setting the head size appropriately in the Transformers allows us to train models with a better performance scaling. We first describe our experimental setup followed by our results and ablation studies on the proposed modifications.

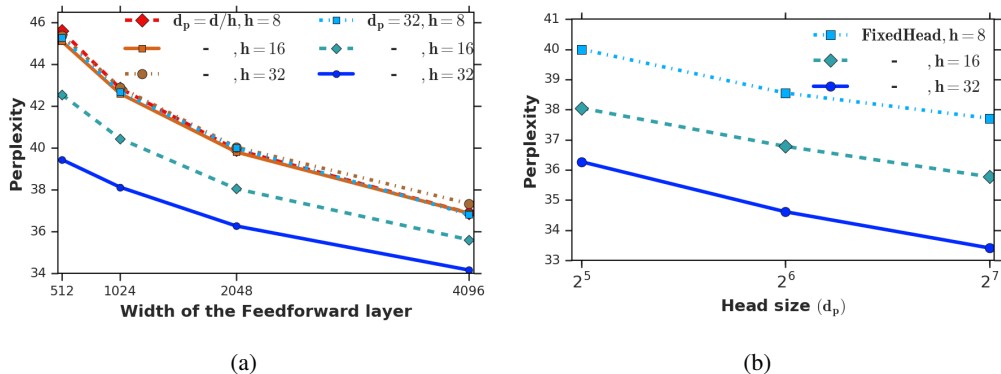

Figure 3: Ablation studies on LM1B: (a) We fix the embedding size of all the models to 256 and vary the capacity of the standard Transformers by increasing the size of the feedforward layers. For the modified models we fix the head size to 32, so 8 head modified model is the same as the 8 head standard Transformer. We notice that again in Transformers increasing the number of heads beyond 16 hurts the performance, whereas with a fixed head size increasing the number of heads monotonically improves the performance. (b) We show the effect of head size on the performance with different number of heads. Both plots clearly show the advantage in having an additional way to tune the capacity of Transformers with a fixed embedding size.

## 4.1 SETUP AND DATASETS

For the language modeling task we use the one billion word benchmark dataset (LM1B) (Chelba et al., 2013). This dataset has around 30M training examples and around 300k examples in the test set. We use a sub-word tokenizer with 32k vocab and cap the input to 256 sequence length. We train a 6 layer Transformer model with the ADAM optimizer using the tensor2tensor library (Vaswani et al., 2018). The detailed experimental setting is presented in Section C.

Multi-Genre Natural Language Inference (MNLI) is a sentence level entailment task, designed to test natural language understanding (Williams et al., 2018). Given a premise sentence and a hypothesis sentence, the goal is to predict whether hypothesis entails, contradicts or is neutral to the premise. We report the classification accuracy for this task. Stanford Question Answering Dataset (SQuAD) is a question answering dataset, where given a paragraph and a question, the goal is to predict the sequence of words in the paragraph that constitute the answer to the question (Rajpurkar et al., 2016). This is a harder word level task, compared to the sentence classification task. We report both Exact Match (EM) and F1 scores for this task. All results in this section are reported on the Dev set, which has not been used in any experimental choices in this paper.

For these latter two tasks, we follow the two stage approach of first pre-training on a language modeling task and then fine-tuning the models on the task data. We follow the same experimental setup for both pre-training and fine-tuning as BERT (Devlin et al., 2018), and use their codebase[1]. We first pre-train our models using the masked language model and the next sentence prediction objectives, and then fine tune the pre-trained model for individual tasks (Devlin et al., 2018). For pre-training we use English Wikipedia and BooksCorpus dataset (Zhu et al., 2015). The input to the models is tokenized using the WordPiece representation with 30000 tokens in the vocabulary. We present the key experiment choices in Section C, and refer the reader to Devlin et al. (2018) for a complete description of the setup.

## 4.2 RESULTS

For our first set of experiments we want to see if the fixed head size Transformer with a smaller embedding size can match the performance of standard Transformers with a larger embedding size. As a baseline for the language modeling task, we train Transformers with the embedding size increasing from 256 to 512 with different number of heads. We train the fixed head size Transformers with a

---

[1]https://github.com/google-research/bert

| # heads | 8 | 12 | 16 | 32 |
|---|---|---|---|---|
| # params | 168M | 193M | 218M | 319M |
| SQuAD - F1 | 89.6±0.17 | 90.25±0.21 | 90.43±0.14 | 90.95±0.14 |
| SQuAD - EM | 82.73±0.21 | 83.18±0.24 | 83.59±0.06 | 84.4±0.29 |
| MNLI | 83.5±0.2 | 84.2±0.2 | 83.9±0.2 | 84.9±0.2 |

(A) Increasing number of heads

| head size | 32 | 64 | 128 | 256 |
|---|---|---|---|---|
| # params | 130M | 142M | 168M | 218M |
| SQuAD - F1 | 88.53±0.06 | 89.51±0.15 | 89.6±0.17 | 90.33±0.23 |
| SQuAD - EM | 81.19±0.21 | 82.41±0.32 | 82.73±0.21 | 83.36±0.48 |
| MNLI | 82.5±0.1 | 83.4±0.3 | 83.5±0.2 | 83.9±0.2 |

(B) Increasing head size

Table 2: (A): 24 layer modified Transformer with a fixed head size of 128 and 512 embedding size shows an improvement in the accuracy with the increasing number of heads. (B) The fixed head size model with 512 embedding size and 8 heads shows an improvement in accuracy with the increasing head size. This shows that indeed head size is an important capacity controlling parameter in the self attention architecture.

fixed embedding size of 256 and a head size of 32, with an increasing number of heads from 4 to 70. We notice that the fixed head size models with an embedding size of 256 can match the performance of standard Transformers with an embedding size of 512 (see Fig. 1). Further this provides a better performance scaling than the standard Transformers. We repeat the similar experiment on the other two tasks, where for baseline we train BERT$_{LARGE}$, a 24 layer, 16 head Transformer architecture, with embedding sizes from 512 to 1024. We compare it with the modified model, with an embedding size of 512 and a head size of 128, with an increasing number of heads from 8 to 32. We again notice that the fixed head size model with 512 embedding size can match the performance of BERT$_{LARGE}$ (see Fig. 2).

Note that simply trying to increase the head size of the standard Transformers by decreasing the number of heads, does not improve the performance, as decreasing the number of heads reduces the expressive power of the model (see Fig. 4). Hence, both the head size and the number of heads needs to be set high enough for better performance.

### 4.3 ABLATION

**Increasing heads.** From Table 1 and Fig. 1a we can see that increasing the number of heads hurts the performance of the Transformer after a certain number. We repeat the same experiments with the fixed head size Transformer, and present the results in Table 2(A) and Fig. 3a. The results show that the performance of the modified model improves monotonically as the number of heads increase. This is because the model capacity (a function of the head size) is no longer reduced with the increasing number of heads.

**Increasing head size.** In Table 2(B) and Fig. 3b, we present comparisons between models with different head sizes. This shows that the gains in the performance of the fixed head size models indeed come from adjusting the head size of the query, key and value layers in the attention unit. The table shows a clear trend of better performance with a larger head size, suggesting that it indeed is an important factor in the performance of the attention models.

## 5 CONCLUSION

In this paper we studied the representation power of the multi-head self attention models and showed that the larger embedding size used in the current models is a consequence of the limitations of the current multi-head attention formulation. We propose a modified way to set the head size that allows us to increase the number of heads without increasing the embedding size. As a consequence we

are able to train Transformers with a smaller embedding size and fewer parameters, without hurting the performance. In the future, it will be interesting to experiment with varying head sizes within an attention block and across layers. This requires further understanding of the role of each layer in computing the context, which is an interesting direction for the future work.

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

## A  Notation

| Embedding size | $d$ |
|:---:|:---:|
| Number of layers | $l$ |
| Number of heads | $h$ |
| Sequence length | $n$ |
| Vocab size | $v$ |
| Head size | $d_p$ |

## B  Proofs

*Proof of Theorem 1.* $\mathbf{d} \geq \mathbf{n}$ **case**. To prove the first part of the result, we present an explicit construction of $\mathbf{W}_k$ and $\mathbf{W}_q$ which allows us to generate $\mathbf{P}$ from $\mathbf{X}$ using the dot product attention. Since $\mathbf{X}$ has full column rank, there exists a left inverse $\mathbf{X}^\dagger = (\mathbf{X}^T\mathbf{X})^{-1}\mathbf{X}^T \in \mathbb{R}^{n \times d}$ such that $\mathbf{X}^\dagger\mathbf{X} = \mathbf{I}_n$. Let $\mathbf{W}_k = \tilde{\mathbf{W}}_k\mathbf{X}^\dagger$ and $\mathbf{W}_q = \tilde{\mathbf{W}}_q\mathbf{X}^\dagger$. Then

$$\mathbf{X}^T\mathbf{W}_k^T\mathbf{W}_q\mathbf{X} = \mathbf{X}^T(\mathbf{X}^\dagger)^T\tilde{\mathbf{W}}_k^T\tilde{\mathbf{W}}_q\mathbf{X}^\dagger\mathbf{X} = \mathbf{I}_n \cdot \tilde{\mathbf{W}}_k^T\tilde{\mathbf{W}}_q \cdot \mathbf{I}_n = \tilde{\mathbf{W}}_k^T\tilde{\mathbf{W}}_q = \tilde{\mathbf{W}}_{kq} \qquad (5)$$

Now that the above choice of $\mathbf{W}_q$ and $\mathbf{W}_k$ has handled the dependence on $\mathbf{X}$, we will choose a $\tilde{\mathbf{W}}_{kq}$ depending on $\mathbf{P}$ and finish the construction. Below we express the Softmax operation on the query and key inner products. Note that the Softmax here is a columnwise operator computing the attention scores for each query. By using (5), we obtain that

$$\text{Softmax}\left[\frac{(\mathbf{W}_k\mathbf{X})^T(\mathbf{W}_q\mathbf{X})}{\sqrt{d_k}}\right] = \text{Softmax}\left[\frac{\tilde{\mathbf{W}}_{kq}}{\sqrt{d_k}}\right] = \exp\left(\frac{\tilde{\mathbf{W}}_{kq}}{\sqrt{d_k}}\right) \cdot \mathbf{D}_{\tilde{\mathbf{W}}_{kq}}^{-1},$$

where $\mathbf{D}_{\tilde{\mathbf{W}}_{kq}}$ is an $n \times n$ diagonal matrix such that

$$(\mathbf{D}_{\tilde{\mathbf{W}}_{kq}})_{ii} = \sum_{j=1}^{n} \exp\left(\frac{(\tilde{\mathbf{W}}_{kq})_{ji}}{\sqrt{d_k}}\right) = \left(\mathbf{1}^T\exp\left(\frac{(\tilde{\mathbf{W}}_{kq})}{\sqrt{d_k}}\right)\right)_i.$$

Hence, we can establish the desired result by showing that there always exists a $\tilde{\mathbf{W}}_{kq}$ that satisfies the following fixed point equation.

$$\exp\left(\frac{\tilde{\mathbf{W}}_{kq}}{\sqrt{d_k}}\right) = \mathbf{P} \cdot \mathbf{D}_{\tilde{\mathbf{W}}_{kq}}. \qquad (6)$$

Given $\mathbf{P}$, to construct such a $\tilde{\mathbf{W}}_{kq}$, we pick an arbitrary positive diagonal matrix $\mathbf{D}_0$, and set

$$\tilde{\mathbf{W}}_{kq} = \sqrt{d_k} \cdot \log(\mathbf{P} \cdot \mathbf{D}_0). \qquad (7)$$

Since $\mathbf{P}$ is a positive matrix, such a $\tilde{\mathbf{W}}_{kq}$ always exists. Next, we verify that this construction indeed satisfies the fixed point equation (cf. (6)). Note that

$$\mathbf{D}_{\tilde{\mathbf{W}}_{kq}} = \text{Diag}\left(\mathbf{1}^T\exp\left(\frac{(\tilde{\mathbf{W}}_{kq})}{\sqrt{d_k}}\right)\right) = \text{Diag}\left(\mathbf{1}^T\mathbf{P}\cdot\mathbf{D}_0\right) = \mathbf{D}_0. \qquad (8)$$

The last equation follows from the fact that $\mathbf{P}$ is a column stochastic matrix. Now, using (7) and (8),

$$\exp\left(\frac{\tilde{\mathbf{W}}_{kq}}{\sqrt{d_k}}\right) = \mathbf{P} \cdot \mathbf{D}_0 = \mathbf{P} \cdot \mathbf{D}_{\tilde{\mathbf{W}}_{kq}}.$$

This completes the first part of the proof.

$\mathbf{d} < \mathbf{n}$ **case**. Consider the case of $d = 1$ and $n = 2$. Then $\mathbf{X} \in \mathbb{R}^{1 \times 2}$ and $\mathbf{W}_q$ and $\mathbf{W}_k \in \mathbb{R}^{1 \times 1}$. Let $\mathbf{X} = [1, 0]$. Then

$$\text{Softmax}\left[\frac{(\mathbf{W}_k\mathbf{X})^T(\mathbf{W}_q\mathbf{X})}{\sqrt{d_k}}\right] = \text{Softmax}\left[\frac{[1,0]^T\mathbf{W}_k^T\mathbf{W}_q[1,0]}{\sqrt{d_k}}\right] = \text{Softmax}\left[\begin{bmatrix}\mathbf{W}_k\mathbf{W}_q & 0 \\ 0 & 0\end{bmatrix}\right].$$

This matrix clearly cannot be used to generate $\mathbf{P}$ that have distinct elements in the second column , e.g., $\mathbf{P} = \begin{bmatrix} 0.5 & 0.75 \\ 0.5 & 0.25 \end{bmatrix}$. $\qquad\square$

*Proof of Theorem 2.* First let us rewrite the MultiHead and FixedMultiHead layers as follows. The MultiHead layer can be rewritten as

$$f_{\mathbf{W}}(\mathbf{X}) = \mathbf{W}_o \cdot \text{MultiHead}(\mathbf{X}) = \sum_{i=1}^{h} \mathbf{W}_o^i \mathbf{W}_v^i \mathbf{X} \cdot \text{Softmax}\left[ (\mathbf{W}_k^i \mathbf{X})^T (\mathbf{W}_q^i \mathbf{X}) / \sqrt{\tfrac{d}{h}} \right],$$

where $\mathbf{W}_o^i$ are $d \times d/h$ matrices and $\mathbf{W}_v^i$, $\mathbf{W}_k^i$, and $\mathbf{W}_q^i$ are $d/h \times d$ matrices. We denote the collection of all parameter matrices as $\mathbf{W}$.

Similarly, rewrite the fixed head size attention layer as

$$g_{\mathbf{V}}(\mathbf{X}) = \mathbf{V}_o \cdot \text{FixedMultiHead}(\mathbf{X}) = \sum_{i=1}^{h} \mathbf{V}_o^i \mathbf{V}_v^i \mathbf{X} \cdot \text{Softmax}\left[ (\mathbf{V}_k^i \mathbf{X})^T (\mathbf{V}_q^i \mathbf{X}) / \sqrt{d_p} \right],$$

where $\mathbf{V}_o^i \in \mathbb{R}^{d \times d_p}$, and $\mathbf{V}_v^i, \mathbf{V}_k^i, \mathbf{V}_q^i \in \mathbb{R}^{d_p \times d}$. Let $\mathbf{V}$ be the collection of all these matrices.

The outline of the proof is basically a case analysis: we divide possible values of $\mathbf{W}$ into three categories, and show in each case that there exists a $\mathbf{X}$ such that $f_{\mathbf{W}}(\mathbf{X}) \neq g_{\mathbf{V}}(\mathbf{X})$. Here are the three cases:

- **Case 1**: $\sum_{i=1}^{h} \mathbf{W}_o^i \mathbf{W}_v^i \neq \sum_{i=1}^{h} \mathbf{V}_o^i \mathbf{V}_v^i$.

- **Case 2**: $\sum_{i=1}^{h} \mathbf{W}_o^i \mathbf{W}_v^i = \sum_{i=1}^{h} \mathbf{V}_o^i \mathbf{V}_v^i$, and there exists $i \in \{1, \ldots, h\}$ such that $\mathbf{U}/\sqrt{d_p} - (\mathbf{W}_k^i)^T (\mathbf{W}_q^i)/\sqrt{d/h}$ is not skew-symmetric.

- **Case 3**: $\sum_{i=1}^{h} \mathbf{W}_o^i \mathbf{W}_v^i = \sum_{i=1}^{h} \mathbf{V}_o^i \mathbf{V}_v^i$, and all $\mathbf{U}/\sqrt{d_p} - (\mathbf{W}_k^i)^T (\mathbf{W}_q^i)/\sqrt{d/h}$ are skew-symmetric.

**Case 1.** In the first case, we can choose any $\mathbf{v}$ such that $(\sum_{i=1}^{h} \mathbf{W}_o^i \mathbf{W}_v^i - \sum_{i=1}^{h} \mathbf{V}_o^i \mathbf{V}_v^i)\mathbf{v} \neq \mathbf{0}$. Choose $\mathbf{X} = \mathbf{v}\mathbf{1}^T = [\mathbf{v} \quad \mathbf{v} \quad \ldots \quad \mathbf{v}]$. Then, note that for any column stochastic matrix $\mathbf{P}$, we have $\mathbf{X}\mathbf{P} = \mathbf{X}$. Therefore,

$$\sum_{i=1}^{h} \mathbf{W}_o^i \mathbf{W}_v^i \mathbf{X} \cdot \text{Softmax}\left[ (\mathbf{W}_k^i \mathbf{X})^T (\mathbf{W}_q^i \mathbf{X}) / \sqrt{d/h} \right] - \sum_{i=1}^{h} \mathbf{V}_o^i \mathbf{V}_v^i \mathbf{X} \cdot \text{Softmax}\left[ (\mathbf{V}_k^i \mathbf{X})^T (\mathbf{V}_q^i \mathbf{X}) / \sqrt{d_p} \right]$$

$$= \sum_{i=1}^{h} \mathbf{W}_o^i \mathbf{W}_v^i \mathbf{X} - \sum_{i=1}^{h} \mathbf{V}_o^i \mathbf{V}_v^i \mathbf{X} = (\sum_{i=1}^{h} \mathbf{W}_o^i \mathbf{W}_v^i - \sum_{i=1}^{h} \mathbf{V}_o^i \mathbf{V}_v^i)\mathbf{v}\mathbf{1}^T \neq \mathbf{0}.$$

**Case 2.** In cases where $\sum_{i=1}^{h} \mathbf{W}_o^i \mathbf{W}_v^i = \sum_{i=1}^{h} \mathbf{V}_o^i \mathbf{V}_v^i$, since $\sum_{i=1}^{h} \mathbf{V}_o^i \mathbf{V}_v^i$ is full rank by assumption and each $\mathbf{W}_o^i \mathbf{W}_v^i$ is at most rank $d/h$, it follows that all columns in $\mathbf{W}_o^i \in \mathbb{R}^{d \times d/h}$ must be linearly independent. Therefore, for any $\mathbf{v} \neq \mathbf{0}$, $\{\mathbf{W}_o^i \mathbf{W}_v^i \mathbf{v}, i = 1, \ldots, h\}$ is a set of linearly independent vectors, because each $\mathbf{W}_o^i \mathbf{W}_v^i \mathbf{v}$ is a linear combination of $d/h$ column vectors of $\mathbf{W}_o^i$ that are linearly independent of other column vectors in $\mathbf{W}_o^j$, $j \neq i$.

Now consider any $\mathbf{v} \in \mathbb{R}^d$, and $\mathbf{X} = \mathbf{v}\mathbf{e}_1^T$, where $\mathbf{e}_1 = (1, 0, \ldots, 0) \in \mathbb{R}^n$. Define $\phi(t) = \exp(t)/(\exp(t) + n - 1)$. Then, we have

$$g_{\mathbf{V}}(\mathbf{X}) = \sum_{i=1}^{h} \mathbf{V}_o^i \mathbf{V}_v^i \mathbf{X} \cdot \text{Softmax}\left[ \mathbf{X}^T \mathbf{U}\mathbf{X} / \sqrt{d_p} \right] = \sum_{i=1}^{h} \mathbf{V}_o^i \mathbf{V}_v^i \mathbf{X} \cdot \text{Softmax} \begin{bmatrix} \frac{\mathbf{v}^T \mathbf{U}\mathbf{v}}{\sqrt{d_p}} & 0 & \cdots & 0 \\ 0 & 0 & \cdots & 0 \\ \vdots & \vdots & \ddots & \vdots \\ 0 & 0 & \cdots & 0 \end{bmatrix}$$

$$= \left( \sum_{i=1}^{h} \mathbf{V}_o^i \mathbf{V}_v^i \right) \left[ \phi\left( \frac{\mathbf{v}^T \mathbf{U}\mathbf{v}}{\sqrt{d_p}} \right) \mathbf{v} \quad \frac{\mathbf{v}}{n} \quad \cdots \quad \frac{\mathbf{v}}{n} \right] = \left( \sum_{i=1}^{h} \mathbf{W}_o^i \mathbf{W}_v^i \right) \left[ \phi\left( \frac{\mathbf{v}^T \mathbf{U}\mathbf{v}}{\sqrt{d_p}} \right) \mathbf{v} \quad \frac{\mathbf{v}}{n} \quad \cdots \quad \frac{\mathbf{v}}{n} \right].$$

Similarly, we can calculate

$$f_{\mathbf{W}}(\mathbf{X}) = \sum_{i=1}^{h} \mathbf{W}_o^i \mathbf{W}_v^i \mathbf{X} \cdot \text{Softmax} \left[ (\mathbf{W}_k^i \mathbf{X})^T (\mathbf{W}_q^i \mathbf{X}) / \sqrt{d/h} \right]$$

$$= \sum_{i=1}^{h} \mathbf{W}_o^i \mathbf{W}_v^i \left[ \phi \left( \frac{\mathbf{v}^T (\mathbf{W}_k^i)^T \mathbf{W}_q^i \mathbf{v}}{\sqrt{d/h}} \right) \mathbf{v} \quad \frac{\mathbf{v}}{n} \quad \cdots \quad \frac{\mathbf{v}}{n} \right].$$

Notice that all the columns of $f_{\mathbf{W}}(\mathbf{X})$ and $g_{\mathbf{V}}(\mathbf{X})$, from the second columns to the last ones, are the same. We now compare the first columns:

$$f_{\mathbf{W}}(\mathbf{X})_{:,1} - g_{\mathbf{V}}(\mathbf{X})_{:,1} = \sum_{i=1}^{h} \left( \phi \left( \frac{\mathbf{v}^T (\mathbf{W}_k^i)^T \mathbf{W}_q^i \mathbf{v}}{\sqrt{d/h}} \right) - \phi \left( \frac{\mathbf{v}^T \mathbf{U} \mathbf{v}}{\sqrt{d_p}} \right) \right) \mathbf{W}_o^i \mathbf{W}_v^i \mathbf{v}.$$

Recall that for any $\mathbf{v} \neq \mathbf{0}$, $\mathbf{W}_o^i \mathbf{W}_v^i \mathbf{v}$ are linearly independent, so $f_{\mathbf{W}}(\mathbf{X})_{:,1} - g_{\mathbf{V}}(\mathbf{X})_{:,1} = \mathbf{0}$ if and only if all $\phi \left( \frac{\mathbf{v}^T (\mathbf{W}_k^i)^T \mathbf{W}_q^i \mathbf{v}}{\sqrt{d/h}} \right) - \phi \left( \frac{\mathbf{v}^T \mathbf{U} \mathbf{v}}{\sqrt{d_p}} \right)$ are zero. However, since there exists $i \in \{1, \ldots, h\}$ such that $\mathbf{U}/\sqrt{d_p} - (\mathbf{W}_k^i)^T (\mathbf{W}_q^i) / \sqrt{d/h}$ is not skew-symmetric, we can choose $\mathbf{v}$ to be one that satisfies $\frac{\mathbf{v}^T (\mathbf{W}_k^i)^T \mathbf{W}_q^i \mathbf{v}}{\sqrt{d/h}} \neq \frac{\mathbf{v}^T \mathbf{U} \mathbf{v}}{\sqrt{d_p}}$, hence making $\phi \left( \frac{\mathbf{v}^T (\mathbf{W}_k^i)^T \mathbf{W}_q^i \mathbf{v}}{\sqrt{d/h}} \right) - \phi \left( \frac{\mathbf{v}^T \mathbf{U} \mathbf{v}}{\sqrt{d_p}} \right) \neq 0$, therefore $f_{\mathbf{W}}(\mathbf{X})_{:,1} - g_{\mathbf{V}}(\mathbf{X})_{:,1} \neq \mathbf{0}$.

**Case 3.** Now consider any $\mathbf{X} = [\mathbf{v}_1 \quad \mathbf{v}_2 \quad \mathbf{0} \quad \ldots \quad \mathbf{0}]$, where $\mathbf{v}_1$ and $\mathbf{v}_2$ will be chosen later. Define $\phi_1(t_1, t_2) = \exp(t_1)/(\exp(t_1) + \exp(t_2) + n - 2)$, $\phi_2(t_1, t_2) = \exp(t_2)/(\exp(t_1) + \exp(t_2) + n - 2)$. Then, we have

$$g_{\mathbf{V}}(\mathbf{X}) = \sum_{i=1}^{h} \mathbf{V}_o^i \mathbf{V}_v^i \mathbf{X} \cdot \text{Softmax} \begin{bmatrix} \frac{\mathbf{v}_1^T \mathbf{U} \mathbf{v}_1}{\sqrt{d_p}} & \frac{\mathbf{v}_1^T \mathbf{U} \mathbf{v}_2}{\sqrt{d_p}} & 0 & \cdots & 0 \\ \frac{\mathbf{v}_2^T \mathbf{U} \mathbf{v}_1}{\sqrt{d_p}} & \frac{\mathbf{v}_2^T \mathbf{U} \mathbf{v}_2}{\sqrt{d_p}} & 0 & \cdots & 0 \\ 0 & 0 & 0 & \cdots & 0 \\ \vdots & \vdots & \vdots & \ddots & \vdots \\ 0 & 0 & 0 & \cdots & 0 \end{bmatrix}.$$

Therefore, the first column of $g_{\mathbf{V}}(\mathbf{X})$ can be written as

$$g_{\mathbf{V}}(\mathbf{X})_{:,1} = \left( \sum_{i=1}^{h} \mathbf{W}_o^i \mathbf{W}_v^i \right) \left[ \phi_1 \left( \frac{\mathbf{v}_1^T \mathbf{U} \mathbf{v}_1}{\sqrt{d_p}}, \frac{\mathbf{v}_2^T \mathbf{U} \mathbf{v}_1}{\sqrt{d_p}} \right) \mathbf{v}_1 + \phi_2 \left( \frac{\mathbf{v}_1^T \mathbf{U} \mathbf{v}_1}{\sqrt{d_p}}, \frac{\mathbf{v}_2^T \mathbf{U} \mathbf{v}_1}{\sqrt{d_p}} \right) \mathbf{v}_2 \right].$$

Similarly, the first column of $f_{\mathbf{W}}(\mathbf{X})$ is

$$f_{\mathbf{W}}(\mathbf{X})_{:,1} = \sum_{i=1}^{h} \mathbf{W}_o^i \mathbf{W}_v^i \left[ \phi_1 \left( \frac{\mathbf{v}_1^T (\mathbf{W}_k^i)^T \mathbf{W}_q^i \mathbf{v}_1}{\sqrt{d/h}}, \frac{\mathbf{v}_2^T (\mathbf{W}_k^i)^T \mathbf{W}_q^i \mathbf{v}_1}{\sqrt{d/h}} \right) \mathbf{v}_1 + \right.$$

$$\left. \phi_2 \left( \frac{\mathbf{v}_1^T (\mathbf{W}_k^i)^T \mathbf{W}_q^i \mathbf{v}_1}{\sqrt{d/h}}, \frac{\mathbf{v}_2^T (\mathbf{W}_k^i)^T \mathbf{W}_q^i \mathbf{v}_1}{\sqrt{d/h}} \right) \mathbf{v}_2 \right].$$

Since $\mathbf{U}/\sqrt{d_p} - (\mathbf{W}_k^1)^T (\mathbf{W}_q^1) / \sqrt{d/h}$ is skew-symmetric by assumption, we have $\mathbf{v}_1^T \left( \frac{\mathbf{U}}{\sqrt{d_p}} - \frac{(\mathbf{W}_k^1)^T (\mathbf{W}_q^1)}{\sqrt{d/h}} \right) \mathbf{v}_1 = 0$ for all $\mathbf{v}_1$. Recall that $\mathbf{U}$ is rank-$d_p$ by assumption, so $\mathbf{U}/\sqrt{d_p} - (\mathbf{W}_k^1)^T (\mathbf{W}_q^1) / \sqrt{d/h}$ is at least rank $d_p - d/h \geq 1$, so we can choose any $\mathbf{v}_1$ such that $\left( \frac{\mathbf{U}}{\sqrt{d_p}} - \frac{(\mathbf{W}_k^1)^T (\mathbf{W}_q^1)}{\sqrt{d/h}} \right) \mathbf{v}_1 \neq \mathbf{0}$.

If both $\frac{\mathbf{U}}{\sqrt{d_p}}\mathbf{v}_1$ and $\frac{(\mathbf{W}_k^1)^T(\mathbf{W}_q^1)}{\sqrt{d/h}}\mathbf{v}_1$ are nonzero, We can always choose $\tilde{\mathbf{v}}_2$ such that $\tilde{\mathbf{v}}_2^T\left(\frac{\mathbf{U}}{\sqrt{d_p}}\right)\mathbf{v}_1 > 0$ and $\tilde{\mathbf{v}}_2^T\left(\frac{(\mathbf{W}_k^1)^T(\mathbf{W}_q^1)}{\sqrt{d/h}}\right)\mathbf{v}_1 < 0$. This means that if we choose $\mathbf{v}_2 = \alpha\tilde{\mathbf{v}}_2$ and scale $\alpha \to \infty$,

$$\phi_1\left(\frac{\mathbf{v}_1^T\mathbf{U}\mathbf{v}_1}{\sqrt{d_p}}, \frac{\mathbf{v}_2^T\mathbf{U}\mathbf{v}_1}{\sqrt{d_p}}\right) \to 0, \quad \phi_2\left(\frac{\mathbf{v}_1^T\mathbf{U}\mathbf{v}_1}{\sqrt{d_p}}, \frac{\mathbf{v}_2^T\mathbf{U}\mathbf{v}_1}{\sqrt{d_p}}\right) \to 1,$$

$$\phi_1\left(\frac{\mathbf{v}_1^T(\mathbf{W}_k^1)^T\mathbf{W}_q^1\mathbf{v}_1}{\sqrt{d/h}}, \frac{\mathbf{v}_2^T(\mathbf{W}_k^1)^T\mathbf{W}_q^1\mathbf{v}_1}{\sqrt{d/h}}\right) \to \frac{\exp(\mathbf{v}_1^T(\mathbf{W}_k^1)^T\mathbf{W}_q^1\mathbf{v}_1/\sqrt{d/h})}{\exp(\mathbf{v}_1^T(\mathbf{W}_k^1)^T\mathbf{W}_q^1\mathbf{v}_1/\sqrt{d/h}) + n - 2},$$

$$\phi_2\left(\frac{\mathbf{v}_1^T(\mathbf{W}_k^1)^T\mathbf{W}_q^1\mathbf{v}_1}{\sqrt{d/h}}, \frac{\mathbf{v}_2^T(\mathbf{W}_k^1)^T\mathbf{W}_q^1\mathbf{v}_1}{\sqrt{d/h}}\right) \to 0.$$

Then, consider the difference $f_\mathbf{W}(\mathbf{X})_{:,1} - g_\mathbf{V}(\mathbf{X})_{:,1}$. Recall that for any $\mathbf{v}$, $\mathbf{W}_o^1\mathbf{W}_v^1\mathbf{v}$ is independent of $\{\mathbf{W}_o^i\mathbf{W}_v^i\mathbf{v}, i \neq 1\}$. This means that, to show $f_\mathbf{W}(\mathbf{X})_{:,1} - g_\mathbf{V}(\mathbf{X})_{:,1} \neq \mathbf{0}$, it suffices to show that

$$\left[\phi_1\left(\frac{\mathbf{v}_1^T(\mathbf{W}_k^1)^T\mathbf{W}_q^1\mathbf{v}_1}{\sqrt{d/h}}, \frac{\mathbf{v}_2^T(\mathbf{W}_k^1)^T\mathbf{W}_q^1\mathbf{v}_1}{\sqrt{d/h}}\right) - \phi_1\left(\frac{\mathbf{v}_1^T\mathbf{U}\mathbf{v}_1}{\sqrt{d_p}}, \frac{\mathbf{v}_2^T\mathbf{U}\mathbf{v}_1}{\sqrt{d_p}}\right)\right]\mathbf{W}_o^1\mathbf{W}_v^1\mathbf{v}_1 +$$

$$\left[\phi_2\left(\frac{\mathbf{v}_1^T(\mathbf{W}_k^1)^T\mathbf{W}_q^1\mathbf{v}_1}{\sqrt{d/h}}, \frac{\mathbf{v}_2^T(\mathbf{W}_k^1)^T\mathbf{W}_q^1\mathbf{v}_1}{\sqrt{d/h}}\right) - \phi_2\left(\frac{\mathbf{v}_1^T\mathbf{U}\mathbf{v}_1}{\sqrt{d_p}}, \frac{\mathbf{v}_2^T\mathbf{U}\mathbf{v}_1}{\sqrt{d_p}}\right)\right]\mathbf{W}_o^1\mathbf{W}_v^1\mathbf{v}_2 \neq \mathbf{0}.$$

If we scale $\mathbf{v}_2 = \alpha\tilde{\mathbf{v}}_2$ with large enough $\alpha$, the second term will dominate the first term and the first term will never be able to cancel the second one. Thus, by choosing large enough $\alpha > 0$, we can make sure that the sum is nonzero.

Even in case where one of $\frac{\mathbf{U}}{\sqrt{d_p}}\mathbf{v}_1$ and $\frac{(\mathbf{W}_k^1)^T(\mathbf{W}_q^1)}{\sqrt{d/h}}\mathbf{v}_1$ is zero (say $\frac{(\mathbf{W}_k^1)^T(\mathbf{W}_q^1)}{\sqrt{d/h}}\mathbf{v}_1 = \mathbf{0}$), we can choose $\tilde{\mathbf{v}}_2 = \frac{\mathbf{U}}{\sqrt{d_p}}\mathbf{v}_1$ and use a similar scaling argument. By choosing large enough $\alpha > 0$ and $\mathbf{v}_2 = \alpha\tilde{\mathbf{v}}_2$, one can show that the difference $f_\mathbf{W}(\mathbf{X})_{:,1} - g_\mathbf{V}(\mathbf{X})_{:,1}$ is nonzero. $\qquad\square$

## C  EXPERIMENTAL SETTINGS

For our experiments with the language modeling (LM1B dataset), we train 6 layer transformer models. We use a batch size of 4096 and train for 250k steps. We use a learning rate of 0.1 with a linear warm up for the first 10k steps. We decay the learning rate with the square root of the number of steps. We train the standard transformers with the embedding dimension varying from 256 to 512. We fix the width of the feed forward layer in the Transformer to be 1024. In addition, we use weight decay of 0.01 and dropout with probability of 0.1 on all the layers.

For our experiments with BERT, we follow the same experimental settings as in (Devlin et al., 2018). We present the key details here and refer the reader to (Devlin et al., 2018). We train with a batch size of 1024 for 450k steps with inputs of sequence length $n = 128$ followed by 50k steps with inputs of sequence length 512. In contrast the BERT paper uses a batch size of 512, and does the pre-training for 900K steps with 128 sequence length inputs and 100k steps with 512 sequence length inputs. We train using ADAM with a learning rate of 1e-4, and a linear warmup and decay schedule as in BERT. We use 5k warmup steps for the first stage, and a re-warmup of 3k steps for the second stage (You et al., 2019). Again, we use weight decay of 0.01 and dropout with probability of 0.1 on all the layers.

For the language modeling task, training is performed on 4 TPUv2 chips for a couple of hours. For BERT models training is performed on 16 TPUv3 chips in the first stage and 64 TPUv3 chips for the second stage. Pre-training with this configuration takes between 2 to 3 days. We did not attempt to find the optimal hyper-parameters for the fixed head size architecture, and use the same hyper-parameters as used for training the standard Transformer.

## D  ADDITIONAL EXPERIMENTAL RESULTS

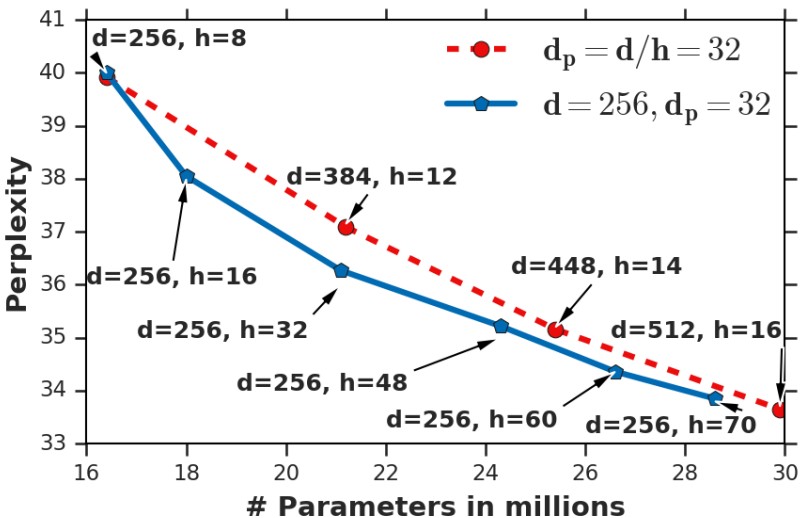

Figure 4: Performance of the standard Transformer training compared with the fixed head size ($d_p$) models for a language modeling task (LM1B) on the test set. Unlike Fig.1, we vary both the embedding size and the number of heads of the standard Transformer to keep its head size fixed to 32. We train the fixed head size models with a fixed embedding size of 256 and a head size of 32, and vary the number of heads from 4 to 70, while matching the number of parameters. The plot again clearly indicates the advantage of the fixed head size models. The main issue with standard Transformers is that fixing the head size to 32, forces the number of heads to be small when the dimension is small. Reducing the number of heads below certain extent hurts the performance of the Transformer.

| # heads | 8 | 12 | 16 | 20 |
|---|---|---|---|---|
| # params | 214M | 252M | 290M | 327M |
| SQuAD - F1 | 90.35±0.14 | 90.48±0.09 | 90.92±0.14 | 90.89±0.08 |
| SQuAD - EM | 83.37±0.12 | 83.67±0.03 | 84.16±0.35 | 84.29±0.16 |
| MNLI | 84.4±0.2 | 84.4±0.2 | 84.7±0.1 | 85.1±0.4 |

(A) Increasing number of heads

Table 3: (A): 24 layer modified Transformer with a fixed head size of 128 and embedding size of 768 shows an improvement in the accuracy with the increasing number of heads.

