# OpenReview forum: "Concise Multi-head Attention Models"
_ICLR.cc/2020/Conference — Reject_

### Official Review · AnonReviewer3 · 2019-10-20
**Official Blind Review #3**

**Rating:** 1

**Review:**

This paper proposes a “concise” version of the well-established Transformer model. The main proposal is to explicitly set the head size in the Transformer model instead of having to divide (share) representation ability amongst heads.

This paper is poorly written and after an entire 2-page long-winded introduction, the reader is left wondering what is the main contribution of this work. The term “concise” is also not well-defined and left vague to readers. I re-read this paper multiple times and the only concluding finding I have is that this paper proposes an explicit way of setting the projection dimension regardless of the number of heads.

After many mathematical formulations, theorems (seemingly ornamental, or handwavy actually), the final contribution seems to be to set the head size of BERT (size of each head) to 128. This is really trivial. The authors kept teasing a “different” way to do this, but this left the reader completely unsatisfied when the different way refers to explicitly setting each head to 128 and using a smaller model overall.

The value 128 is derived from a theorem derived by the authors, which suggests that each head should at least be greater or equal than the sequence length (the sequence length here stated by the authors is 128). I’m not very convinced by the argument. While it is intuitive that each head has to be sufficiently large, being under-sized can be made up for with multiple heads. It is also not clear why every X and P must be expressed with transforms W_q and W_k. P here represents the affinity matrix between tokens in a sequence.  It does not make any sense to me to ensure that every variation of P can be expressed because P is literally the pairwise scores between every token in the fully-connected attention graph.

While I did not have the luxury of time to parse the Appendix to validate the legitimacy of the proof, I think the overall shortcomings of the paper (highly non-readable, bad presentation and perhaps a fair attempt at masking the lack of contribution) warrants a clear reject from me.


**Experience Assessment:**

I have published in this field for several years.

**Review Assessment: Checking Correctness Of Derivations And Theory:**

I assessed the sensibility of the derivations and theory.

**Review Assessment: Checking Correctness Of Experiments:**

I assessed the sensibility of the experiments.

**Review Assessment: Thoroughness In Paper Reading:**

I read the paper at least twice and used my best judgement in assessing the paper.

---

> ### Author Response · Authors · 2019-11-09
> **Response to Review #3**
>
> Comment:  “... what is the main contribution of this work”
> Response:  In this paper we identify and highlight a key capacity bottleneck in multihead attention. We rigorously analyze this by proving capacity bounds of attention as a function of the head size. Based on this analysis, we propose a principled solution of using fixed head size attention layers to alleviate this bottleneck, and experimentally show superior performance (smaller embedding size, better parameter scaling) on 3 standard NLP tasks.
>
> We clearly state this in the abstract “As a solution, we propose a new way to set the projection size in attention heads.. ,”. We again state it in the introduction, contributions - point 2: “We propose a new way to set the head size, and show the proposed fixed head size layers are strictly better than the standard multi-head attention layers...”
>
> Comment: “...theorems (seemingly ornamental, or handwavy actually)...”
> Response: This is an unfair characterization of our work. Theorem 1 shows the relationship between head size and the representation power of a single attention head. Theorem 2 shows that the FixedMultiHead has strictly more representative power than MultiHead layer. We present both the theorems with precise mathematical statements and complete proofs. We are happy to update the paper, if the reviewer noticed an issue with a particular statement, or a particular step in the proof.
>
> Comment:  “...the different way refers to explicitly setting each head to 128…”
> Response: As we mentioned earlier, we clearly state both in the abstract and the introduction, that we train a fixed head size model, as opposed to using the d/h heuristic. We emphasize that our contribution is not just setting the head size parameter, but rather an analysis of the capacity of the multihead attention, and consequently a principled alternative to the current heuristic used to set the head size. Our experiments clearly indicate the advantage of using fixed head size in improving transformer training. We are able to train models with a smaller embedding size (512 vs 1024 for BERT) and with better parameter scaling.
>
> Comments:  “...being under-sized can be made up for with multiple heads…”
> Response: We show in Theorem 2 that this is not the case. MultiHead has strictly lower expressive power than the Fixed MultiHead. Intuitively, increasing the number of heads decreases the head size, and negatively affects the capacity if the head size becomes too small. We establish this both theoretically and empirically.
>
> Comments: “...overall shortcomings of the paper...”
> Response: This is again a wrong characterization of the paper. We ask the reviewer to kindly state their specific issues with the presentation.
>
> We ask the reviewer to kindly reconsider their score based on the importance of our analysis and our experimental contributions described above.

---

### Official Review · AnonReviewer2 · 2019-10-22
**Official Blind Review #2**

**Rating:** 3

**Review:**

This work studies the head size <--> head number tradeoff in multihead attention. It argues and formally establishes that (1) the expressivity of an attention head is determined by its dimension and (b) fixing the head dimension, one gains additional expressive power by using more heads. In response to such observations, the paper proposes Fixed Multihead Attention, where the constraint that `head_size * number_of_heads = embedding_size` in standard multihead attention is lifted; and it allows for using more attention heads without making each head smaller. One can control the total amount of parameters by using smaller embedding sizes, making it comparable (in terms of #parameters) to standard multihead attention. Empirical results on language modeling and NLI tasks confirms the arguments.

Pros:
- The arguments on head size and head number tradeoff could be inspiring to future works.
- A simple approach that proves strong in several NLP tasks.

Cons:
- The theoretical discussion imposes too strong assumptions that might make it less interesting in practice.
- No NMT experiments.
- The takeaway seems a bit trivial.

Details:
- Theorem 1 presents a rank-based view of each attention head's capacity, which is nice. Yet it is still unclear whether it is the case that the more expressive are the heads the better. For example, several recent works argues for specialized attention heads, i.e., each head has specific "job," which may not require it being very expressive [1, 2]. Further, other works shows that a low-rank P matrix could be beneficial [3, 4, 5], which contradicts the argument in this work. It would be nice if the authors and discuss this in the revision

(To be clear, I do believe this is still an open question, and do not think presenting a different view from previous works hurts the contribution of this work in any way.)

- Theorem 2. I didn't carefully check the proof. Why is it required that the V matrices for each head have the same product. For both Thm.1 and 2, it would be nice to see some discussion on how they translate into the models in practice.

- Can the authors compare the training/inference speed? It probably will be the same as standard transformers, but it would be nice to confirm.

- Figure 1: the caption says trying out embedding sizes from 256 to 512. But it seems that only 4 values are tried. Can the authors comment on this? Also, it is a bit awkward to plot a line chart out of 4 points. Same for Figure 2.

- It would be nice to see some NMT experiments.

- The proposed method is so straightforward that I'm actually very surprised that this paper is the first trying this. The authors might need justify the technical contribution more.

(I'm on the fence for this one, but the system doesn't allow me to. I'm happy to revise the score if the authors can address my concerns.)


[1] Analyzing Multi-Head Self-Attention: Specialized Heads Do the Heavy Lifting, the Rest Can Be Pruned. https://arxiv.org/abs/1905.09418

[2] Are Sixteen Heads Really Better than One? https://arxiv.org/abs/1905.10650

[3] Generating Long Sequences with Sparse Transformers. https://arxiv.org/abs/1904.10509

[4] Generating Long Sequences with Sparse Transformers. https://arxiv.org/pdf/1904.10509.pdf.

[5] Adaptively Sparse Transformers. https://arxiv.org/abs/1909.00015.

**Experience Assessment:**

I have published in this field for several years.

**Review Assessment: Checking Correctness Of Derivations And Theory:**

I assessed the sensibility of the derivations and theory.

**Review Assessment: Checking Correctness Of Experiments:**

I assessed the sensibility of the experiments.

**Review Assessment: Thoroughness In Paper Reading:**

I read the paper at least twice and used my best judgement in assessing the paper.

---

> ### Author Response · Authors · 2019-11-09
> **Response to Review #2**
>
> We thank the reviewer for the detailed assessment of our paper. Below we address their concerns, starting with their last question.
>
> Comment: “...I'm actually very surprised that this paper is the first trying this.The authors might need justify the technical contribution more.”
> Response: To reiterate our contribution, In this paper we identify and highlight a key capacity bottleneck in multihead attention. We rigorously analyze this by proving capacity bounds of attention as a function of the head size. Based on this analysis, we propose a principled solution to alleviate this bottleneck, and experimentally show superior performance (smaller embedding size, better parameter scaling) on 3 standard NLP tasks.
>
> The main technical complexity is in analyzing if the low rank capacity bottleneck due to a decrease in head size is alleviated by the increase in number of heads, in the MultiHead. We show in Theorem 2 that MultiHead indeed suffers in representation power because of the rank constraint, whereas Fixed MultiHead does not. We note that our argument used to prove this result relies on a novel construction and constitutes the main technical contribution of the paper.  We agree that the proposed change to the Transformer training is a relatively simple change, which is actually an advantage of the approach considered.
>
> Comment:  “other works … It would be nice if the authors and discuss this in the revision”
> Response: We thank the reviewer for sharing these papers. We updated our related works section. [1, 2] study the importance of different heads in an attention layer. They observe that, during inference, many of the heads in each layer can be pruned away with a little effect on the prediction. However, this approach still requires multiple heads during the training.
>
> Low rank/sparse P: We first note that fixing the head size does not disallow low rank P, rather it gives the model capacity to learn arbitrary P as required by the task. Fixing the head size also gives us the ability to choose a smaller embedding size, as we have more degrees of freedom in designing the architecture.
>
> [3-5] impose a sparsity structure on the attention layer during training to improve both interpretability and performance. Fixing the head size will in fact make it easier to learn such sparsity patterns, as a low rank constraint does not allow a head to express all possible sparsity patterns. Combining these techniques can hence potentially enable training of sparse attention models with a smaller embedding size.
>
> Comment: “Theorem 2 … Why is it required that the V matrices for each head have the same product?”
> Response: In Theorem 2 we show that the standard MultiHead layer has strictly smaller representation power compared to the Fixed MultiHead layer. We prove it by constructing examples of functions that can be represented by the Fixed MultiHead layer, but the standard MultiHead layer fails to represent. We set the product of V matrices from different heads to be the same, only to keep the proof simple and easy to read. We can generalize the construction to different V’s for each head at the cost of more notation in the proof.
>
> Comment:  “Can the authors compare the training/inference speed? It probably will be the same as standard transformers...”
> Response: The inference time, measured for a few of the trained models, shows correlation with the parameter scaling, and is similar to the standard Transformers. We will include the complete statistics in the final version.
>
> Comment: “Figure 1: the caption says trying out embedding sizes from 256 to 512…”
> Response: For training Transformers with the head size heuristic d_p = d/h, we are forced to choose d to be a multiple of h, this limits us to only certain choices of d. For the baseline experiments, in addition to varying the embedding dimension, we also repeat the experiments with 3 different choices of heads [8,16, 32], 4 different choices of width [512, 1024, 2048, 4096] (Fig. 3). We also are limited in our computation budget, as we also present experimental results (Fig. 2) with ablations (Table 2) for the larger/expensive BERT setup.
>
> Comment: “It would be nice to see some NMT experiments.”
> Response: We evaluated the proposed fixed head size setting on language modeling (LM1B) and BERT experiments, with fine tuning on MNLI and SQuAD datasets. We chose these tasks as they are encoder only models, and allows us to test at different scales. Testing the changes in an encoder-decoder framework, for NMT, is an interesting question, which we plan to pursue in the future.

---

### Official Review · AnonReviewer1 · 2019-10-25
**Official Blind Review #1**

**Rating:** 3

**Review:**

This work discusses how to set the projection size for each head (head size) in multi-head attention module, especially Transformer. Theorem 1 is interesting, which points out a lower bound for the head size. The proposed method is to decouple the dependency between the head size and the embedding size. The experiments show that the proposed method is able to achieve comparable performance to BERT with fewer training cost.

The lower bound for the head size is a valuable result. However, the novelty is very limited. To decouple the dependency between the head size and the embedding size is not a novel point. In BERT/Transformer, it is set d_q=d_k=d_v=d, which is not a strict constraint. The only constraint in attention is to have d_q=d_k to allow dot product. Therefore, the proposed method is more like a tuning of hyper-parameters.

**Experience Assessment:**

I have published one or two papers in this area.

**Review Assessment: Checking Correctness Of Derivations And Theory:**

I assessed the sensibility of the derivations and theory.

**Review Assessment: Checking Correctness Of Experiments:**

I assessed the sensibility of the experiments.

**Review Assessment: Thoroughness In Paper Reading:**

I read the paper at least twice and used my best judgement in assessing the paper.

---

> ### Author Response · Authors · 2019-11-09
> **Response to Review #1**
>
> Comment:  “To decouple the dependency between the head size and the embedding size is not a novel point, ... is not a strict constraint.”
>
> Response: In this paper we identify and highlight a key capacity bottleneck in multihead attention. We rigorously analyze this by proving capacity bounds of attention as a function of the head size. Based on this analysis, we propose a principled solution to alleviate this bottleneck, and experimentally show superior performance (smaller embedding size, better parameter scaling) on 3 standard NLP tasks.
>
> Calling our contribution as  “...tuning of hyper-parameters” is an unfair characterization that ignores our analysis, and our principled approach for either reducing the embedding size or improving the model performance for the same embedding size by addressing the bottleneck of head size.
>
> We disagree with the reviewer on this and emphasize that for the training of transformers, this is indeed a novel way to set the head size. All the existing works on transformers such as GPT, BERT, RoBERTa etc., following the initial paper, use this heuristic of setting the head size to d/h, and there has been no work on questioning the optimal way to set the head size. Thus, the quest to improve the model performance via increasing the number of heads has necessitated large embedding sizes of these models, which we show can be reduced by setting the head size appropriately. Our work is the first such paper to analyze how the head size affects the model capacity and show a principled way to set it.  Our ablation studies clearly show the importance of setting the head size the right way, to train Transformers with a smaller embedding size.
>
> There have been only a few works studying the capacity of transformers, and given their popularity, we believe our work will serve as a good first step towards their better understanding in the future.
>
> We ask the reviewer to kindly reassess the paper.

---

### Decision · Program_Chairs · 2019-12-19

**Decision:**

Reject

**Comment:**

This paper studies tradeoffs in the design of attention-based architectures. It argues and formally establishes that the expressivity of an attention head is determined by its dimension and that fixing the head dimension, one gains additional expressive power by using more heads.

Reviewers were generally positive about the question under study here, but raised important concerns about the significance of the results and the take-home message in the current manuscript. The AC shares these concerns, and recommends rejection, while encouraging the authors to address the concerns raised during this discussion.